# Functional renormalization group
# for non-Hermitian and $\mathcal{PT}$-symmetric systems

**Lukas Grunwald[1][*], Volker Meden[1] and Dante M. Kennes[1,2]**

**1** Institut für Theorie der Statistischen Physik, RWTH Aachen University,
52056 Aachen, Germany
**2** Max Planck Institute for the Structure and Dynamics of Matter,
22761 Hamburg, Germany

[*] Lukas.Grunwald@rwth-aachen.de

## Abstract

We generalize the vertex expansion approach of the functional renormalization group to non-Hermitian systems. As certain anomalous expectation values might not vanish, additional terms as compared to the Hermitian case can appear in the flow equations. We investigate the merits and shortcomings of the vertex expansion for non-Hermitian systems by considering an exactly solvable $\mathcal{PT}$-symmetric non-linear toy-model and reveal, that in this model, the fidelity of the vertex expansion in a perturbatively motivated truncation schema is comparable with that of the Hermitian case. The vertex expansion appears to be a viable method for studying correlation effects in non-Hermitian systems.



# 1  Introduction

The hermiticity of a Hamiltonian is one of the key postulates of quantum mechanics, that guarantees reality of eigenvalues and a unitary time evolution. However, in a variety of fields of modern quantum physics, non-Hermitian (nH) Hamiltonians are routinely used as an effective description for the non-conserving dynamics in open systems [1,2]. For instance, the imaginary part of the eigenvalues describes the gain and loss structures of the underlying system.

Moreover, hermiticity is only a sufficient, but not a necessary condition for the reality of eigenenergies and the existence of a unitary time evolution. In 1998 Bender *et.al.* [3] showed numerically that the class of nH Hamiltonians $H = p^2 + x^2 + (ix)^\epsilon$ with parameters $\epsilon \geq 2$ have an entirely real spectrum. Three years later this was also confirmed analytically [4]. These features were related to the Hamiltonian conserving a symmetry composed of parity ($\mathcal{P}$) and time reversal ($\mathcal{T}$), so called parity-time ($\mathcal{PT}$) symmetry. It was later realized that by varying parameters in nH $\mathcal{PT}$-symmetric Hamiltonians one can find critical points (so called exceptional points [5]) where the system transition from having an entirely real to a complex spectrum. The complex eigenvalues appear in complex conjugate pairs [6]. This is often called $\mathcal{PT}$-symmetry breaking, because the eigenstates of the Hamiltonian are no longer eigenstates of $\mathcal{PT}$.

Depending on the community, $\mathcal{PT}$-symmetric Hamiltonians are seen as a fundamental extension of quantum mechanics [6] or as an effective description of nH systems having balanced loss and gain [1,2]. Irrespective of the viewpoint, nH (and in particular $\mathcal{PT}$-symmetric) models show a plethora of novel effects, ranging from new topological features like the nH skin-effect [7], directional invisibility in optical systems [8] to new twists in traditional models such as the resonant level model [9]. Much of the attention so far has been on non-interacting and matrix-models, without any correlations. But just like in the Hermitian case, one can expect that correlations drastically modify the physics in nH systems and even lead to completely new effects. This has also been observed in the few studies that focused on correlated nH systems, see e.g. [10–14].

The proper treatment of correlations is challenging and often requires advanced theoretical tools. One of those tools, successfully employed in the Hermitian case is the functional renormalization group (FRG) [15]. The FRG can be derived in the path integral formulation of quantum many-body theory where one introduces a flow parameter $\lambda$ into the action, such that at $\lambda = \lambda_\infty$ the theory becomes exactly solvable while for $\lambda = \lambda_0$ one recovers the original theory. One can then derive an exact functional-differential equation for the effective action in $\lambda$, the so called Wetterich equation [15], which can be used to integrate the exact solution from $\lambda_\infty$ to $\lambda_0$, where it corresponds to the solution of the original theory. In practice the Wetterich equation is too complicated to be solved exactly such that various approximation schemes were developed, the mainstreams being derivative and vertex expansion [16–20]. The FRG has already been applied to nH systems in the context of QFT in a derivative expansion [21], but the other common treatment, the vertex expansion (VE), heavily used in (non-relativistic) quantum many-body theory [16,19], was so far not employed.

In this work we aim to generalize the FRG-VE to nH systems. We will test the fidelity of FRG-VE for nH systems by considering an exactly solvable $\mathcal{PT}$-symmetric toy-model. There

we will show that the FRG-VE provides an approximation whose fidelity is comparable to the Hermitian case. In particular, we consider one of the simplest $\mathcal{PT}$-symmetric models with a non-linearity, being part of the class studied in [3] and defined by

$$H = \frac{p^2}{2} + \frac{x^2}{2} + \frac{ik}{3!}x^3 \, , \tag{1}$$

with $k \in \mathbb{R}^+$ and $i$ denoting the imaginary unit, at temperature T = 0. The non-linearity $ikx^3$ in the single-particle model can be viewed as a proxy for interactions/correlations in a many-body context. This model was already analyzed with perturbation theory [22] and with the FRG in a derivative expansion [21, 23]. The latter two papers also discuss the renormalization group for $\mathcal{PT}$-symmetric systems in a wider context.

We begin by presenting the model in more detail in section 2. Next to the $\mathcal{PT}$-symmetry, the non-vanishing vacuum expectation value $\langle x \rangle \neq 0$ is a defining feature of this model, that necessitates some modifications in the FRG [24] which we describe in section 3. We then discuss the corresponding classical integral and the quantum theory in section 4. We conclude with an outlook and an overview of future projects.

## 2 Models

In this work we study the non Hermitian (nH) $\mathcal{PT}$-symmetric toy-Hamiltonian defined by (1), which is part of the class introduced in [3]. The Hamiltonian describes a harmonic oscillator coupled to a gain-loss structure, represented by $ikx^3$: For $x > 0$ the system gains probability whilst for $x < 0$ it looses it at the same rate. Next to the quantum theory defined by (1) we will also consider the corresponding classical counterpart (see below for a precise definition).

This model is non-Hermitian (nH), but has a $\mathcal{PT}$-symmetry which leads to a completely real spectrum as was shown numerically [3] and analytically [4]. In general, $\mathcal{PT}$-symmetry is not sufficient for the reality of the spectrum, there being the effect of $\mathcal{PT}$-symmetry breaking [6, 25], where even though the Hamiltonian has a $\mathcal{PT}$-symmetry, the eigenfunctions of $H$ are not eigenfunctions of $\mathcal{PT}$. For (1) this complication does not arise [3, 4]. We thus only return to $\mathcal{PT}$-symmetry breaking in the final section of this paper. The Hamiltonian (1) describes one of the simplest $\mathcal{PT}$-symmetric systems containing a non-linearity and is thus a perfect test bed for a first application of functional renormlaization group in a vertex expansion (FRG-VE) comparable to the quartic anharmonic oscillator in the Hermitian case [20, 26].

Let us first comment on the structure of the single particle Hilbert space [25, 27–29]: Assume we have a general (nH) Hamiltonian $H$ with a complete biorthonormal eigenbasis of right and left eigenstates $\{|R_\alpha\rangle, |L_\alpha\rangle\}$ defined by the Schrödinger equation

$$i\partial_t |R_\alpha\rangle = H |R_\alpha\rangle = E_\alpha |R_\alpha\rangle \, , \tag{2}$$

$$i\partial_t |L_\alpha\rangle = H^\dagger |L_\alpha\rangle = E_\alpha^* |L_\alpha\rangle \, , \tag{3}$$

with $\langle L_\alpha | R_\beta \rangle = \delta_{\alpha\beta}$. For notational convenience we assumed that the eigenvalues $E_\alpha$ are not degenerate.[1] The postulates of quantum mechanics demand the Hilbert space to be equipped with a time-independent and positive-definite norm. This can be achieved by introducing a metric-operator $V$, that in the case of a real spectrum is given by $V = \sum_\alpha |L_\alpha\rangle\langle L_\alpha|$ and acts as

$$VHV^{-1} = H^\dagger . \tag{4}$$

---

[1]For the Hamiltonian (1) the eigenvalues are indeed not degenerate.

The metric naturally leads to a scalar product and a completeness relation defined by

$$\langle R_\alpha | V | R_\beta \rangle = \langle L_\alpha | R_\beta \rangle = \delta_{\alpha\beta}, \tag{5}$$

$$\sum_\alpha |R_\alpha\rangle\langle L_\alpha| = \sum_\alpha |L_\alpha\rangle\langle R_\alpha| = 1. \tag{6}$$

Here we see that the main purpose of the $V$-operator is to distinguish left and right eigenvectors of $H$. The scalar product defined by (5) induces a positive definite and time independent norm (use (4)). In the case of a Hermitian Hamiltonian it coincides with the regular Dirac-scalar product and thus presents a natural generalization to nH systems.

All expectation values and overlaps have to be calculated in the $V$-scalar product, but this is no hindrance to generating the matrix representation of an operator $A$. For the Hamiltonian (1) we can for example consider the occupation number basis of the harmonic oscillator $\{|n\rangle\}$. Expectation values (and energies) can then be evaluated as

$$\langle L_\alpha | A | R_\beta \rangle = \sum_{nm} \langle L_\alpha | n \rangle \, \langle n | A | m \rangle \, \langle m | R_\beta \rangle, \tag{7}$$

where $\langle n | L_\alpha \rangle$ and $\langle m | R_b \rangle$ are left/right eigenvectors obtained from the matrix representation of $H$.

Path integrals can naturally be generalized to nH and $\mathcal{PT}$-symmetric systems [30, 31] as long as their biorthogonal basis is complete, that is (6) holds. Starting from the canonical partition function with $\beta = \frac{1}{T}$, one can write

$$Z = \mathrm{tr}\left(e^{-\beta H}\right) = \sum_n \langle L_n | e^{-\beta H} | R_n \rangle = \int \mathrm{d}x \, \langle x | e^{-\beta H} | x \rangle = \int_{x(0)=x(\beta)} \mathcal{D}x \, e^{-S(x)},$$

where for the second to last equal sign we used (6) and in last step employed the standard slicing derivation [32]. Eventually we are interested in the $\beta \to \infty$ limit. The imaginary time action $S(x)$ for our model reads

$$S(x) = \int_0^\beta \mathrm{d}\tau \left\{ \frac{\left(\partial_\tau x(\tau)\right)^2}{2} + \frac{x(\tau)^2}{2} + \frac{ik}{3!} x(\tau)^3 \right\}. \tag{8}$$

For further calculations it is convenient to switch to Matsubara-frequency space with frequencies $\omega_n = \frac{2\pi}{\beta} n$, $n \in \mathbb{Z}$ yielding

$$S(x) = \frac{1}{2} \sum_{\omega_n} x(-\omega_n) \left[\mathcal{G}_0(\omega_n)\right]^{-1} x(\omega_n) + \frac{ik}{3!\sqrt{\beta}} \sum_{\omega_1,\ldots,\omega_4} \delta_{\sum \omega_i, 0} \, x(\omega_1) x(\omega_2) x(\omega_3) x(\omega_4)$$

$$= S_0(x) + S_{\mathrm{int}}(x), \tag{9}$$

where we have defined the free propagator $\left[\mathcal{G}_0(\omega_n)\right]^{-1} = \left(\omega_n^2 + 1\right)$ and split the action into a free part $S_0(x)$ and an interaction/non-linearity $S_{\mathrm{int}}(x)$. Adding a source term coupling to $x(\omega)$ we define the generating functional for Greens functions as

$$Z(j) = \frac{1}{Z_0} \int \mathcal{D}x \, e^{-S_0(x) - S_{\mathrm{int}}(x) + (j,x)}, \tag{10}$$

where we introduced the shorthand $(j,x) = \sum_\omega j(\omega) x(\omega)$ [2] and normalized by the free theory $Z_0 = Z(0)|_{k=0}$. This will be the starting point for FRG and perturbation theory in the quantum-mechanical model.

---

[2] The plus-sign in *both* arguments is intentional and chosen for notational convenience. If we would include the source-term as $\sum_\omega j(-\omega) x(\omega)$, this would only lead to several minus-signs in definitions and intermediate steps of the derivation.

Before studying the full-quantum theory defined by (10) we consider the corresponding classical integral, which we obtain by only keeping the $\omega_n = 0$ term and setting $\beta = 1$. The corresponding generating function is defined as

$$Z(j) = \frac{1}{Z_0} \int_{-\infty}^{\infty} dx \; e^{-\frac{1}{2\mathcal{G}_0}x^2 - \frac{ik}{3!}x^3 + jx} \tag{11}$$

with normalization $Z_0 = \sqrt{2\pi\mathcal{G}_0}$. Here we kept a general $\mathcal{G}_0 > 0$ for convenience, which (by comparison with (1)) will be set to 1 at the end. The 'theory' described by (11) is analytically solvable and thus presents a convenient first test for the FRG-VE and other methods which we will outline in the next section.

# 3 Methods

## 3.1 FRG with Vacuum Expectation Values

In several publications the FRG is introduced in a didactic manner [16–20, 26]. Here we will give a very brief outline of the derivation and focus on the modifications necessary because of $\langle x \rangle \neq 0$ when considering the Hamiltonian (1). We derive slightly modified equations compared to those introduced in [24] where the situation with $\langle x \rangle \neq 0$ was already discussed in the context of the vertex-expansion for Hermitian systems (see below).

Our starting point is the cumulant generating functional for an action in Matsubara-space which generates the connected Greens functions $G_n^c$

$$W(j) = \log\big(Z(j)\big) = \log\left(\frac{1}{Z_0} \int \mathcal{D}x \; e^{-S_0(x) - S_{\text{int}}(x) + (j,x)}\right), \tag{12}$$

$$G_{\omega_1,\dots,\omega_n}^{(n),c} = \frac{\delta^n W(j)}{\delta j(\omega_1)\dots j(\omega_n)}\bigg|_{j=0}, \tag{13}$$

where we use the shorthand notation $(j, A\phi) = \sum_{\omega \nu} j(\nu)A(\nu, \omega)\phi(\omega)$. We have split the action into a Gaussian part $S_0(x) = \frac{1}{2}\big(x, \mathcal{G}_0^{-1}x\big)$ with free propagator $\mathcal{G}_0$ and an interaction/non-linearity $S_{\text{int}}(x)$. The vertex generating functional is defined as a Legendre transform of $W(j)$ and generates the vertex functions $\Gamma^{(n)}$

$$\Gamma(\phi) = (j, \phi) - W(j) - \frac{1}{2}\big(\phi, [G_0]^{-1}\phi\big), \tag{14}$$

$$\phi(\omega) = \frac{\partial W(j)}{\partial j(\omega)} = \langle x(\omega) \rangle_j, \tag{15}$$

$$\Gamma_{\omega_1,\dots,\omega_n}^{(n)} = \frac{\delta^n \Gamma(\phi)}{\delta\phi(\omega_1)\dots\phi(\omega_n)}\bigg|_{\phi=\phi^\star}, \tag{16}$$

where $\phi^\star = \langle x \rangle|_{j=0}$.[3] One can derive general relations between the vertex functions and connected Greens functions [32], but only $G^{(2),c} = \big(\mathcal{G}_0^{-1} + \Gamma^{(2)}\big)^{-1}$ is relevant at this point.

To set up the FRG for computing the vertex function (and from this observables and correlation functions) we introduce a cutoff $\lambda$ into the free propagator $\mathcal{G}_0 \to \mathcal{G}_0^\lambda$. Calculating the

---

[3]Note that the $\star$ does not indicate complex conjugation, but is rather a part of the symbol $\phi^\star$.

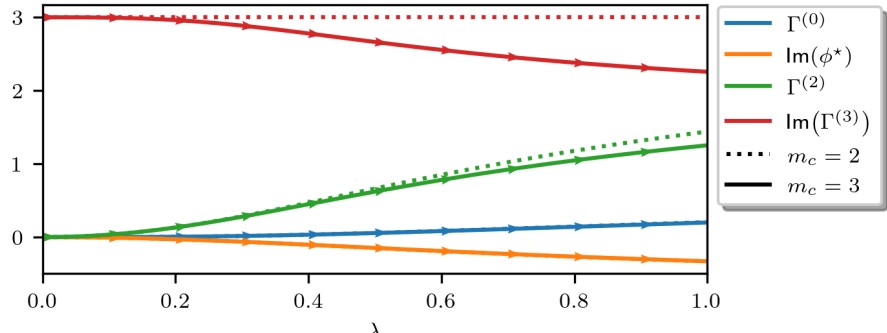

Figure 1: FRG flow of the vertex functions at $k = 3$ for cutoff function $f_\lambda(\omega) = \lambda$ in the quantum case. Full lines show $m_c = 3$ and dotted lines $m_c = 2$ results. For frequency dependent quantities we show the $\omega = 0$ value. Notice in particular the non-trivial flow of $\phi_\lambda^\star$ which necessitates to keep $\phi_\lambda^\star \neq 0$ in the FRG treatment.

$\partial_\lambda$-derivative of (14) we arrive at the (functional-differential) Wetterich equation [15]

$$\partial_\lambda \Gamma^\lambda = \partial_\lambda \log(Z_0^\lambda) + \frac{1}{2} \mathrm{tr}\left( Q_\lambda \left[ \frac{\delta^2 \Gamma^\lambda}{\delta \phi^2} + \left[ \mathcal{G}_0^\lambda \right]^{-1} \right]^{-1} \right), \tag{17}$$

$$\lim_{\lambda \to \lambda_\infty} \Gamma^\lambda(\phi) = S_{\mathrm{int}}(\phi), \tag{18}$$

where we introduced the shorthand $Q_\lambda = \partial_\lambda \left[ \mathcal{G}_0^\lambda \right]^{-1}$. The third term in (14) is needed in order to generate well defined initial conditions for the FRG-cutoff scheme we use: We introduce the cutoff as a multiplicative regulator $\mathcal{G}_0^\lambda(\omega) = \mathcal{G}(\omega) f_\lambda(\omega)$ such that $\mathcal{G}_0^{\lambda_\infty} = 0$ and $\mathcal{G}_0^{\lambda_0} = \mathcal{G}_0$.

In the vertex expansion one expands $\Gamma(\phi)$ in a Taylor-series whose coefficients are the vertex functions $\Gamma^{(n)}$, viz

$$\Gamma(\phi) = \sum_{n=0}^\infty \frac{1}{n!} \sum_{i_1 \dots i_n} \Gamma^{(n)}_{i_1, \dots, i_n} \delta\phi(i_1) \dots \delta\phi(i_n), \quad \text{with} \quad \delta\phi(i) = \phi(i) - \phi_\lambda^\star \delta_{i,0}. \tag{19}$$

The expansion point for this Taylor series is $\phi_\lambda^\star(\omega) = \langle x(\omega) \rangle |_{j=0}$ (see (16)), which is 0 in conventional condensed matter systems without superfluidity. Here $\phi_\lambda^\star(\omega) = \delta_{\omega,0} \phi_\lambda^\star \neq 0$ as can be already seen by inspecting the action (9). Since $\phi_\lambda^\star$ is not known beforehand and is a macroscopic observable, it will have a non trivial RG-flow [24], as we show by the orange line in figure 1 ($\lambda_\infty = 0, \lambda_0 = 1$). The calculation leading to this figure and the precise meaning of $m_c$ will be explained below.

The flow equations for the vertex functions $\Gamma^{(n)}$ can be obtained by inserting the Taylor expansion (19) into (17) and comparing coefficients. This generates a hierarchy of differential equations for the vertex functions, in which the equation for $\Gamma^{(n)}$ depends on $\Gamma^{(n+1)}$ and $\Gamma^{(n+2)}$. To close this system and thus make it numerically tractable, we use a perturbatively motivated truncation: In the flow equation for $\Gamma^{(m_c)}$ we set $\Gamma^{(m>m_c)} = \Gamma^{(m>m_c)}|_{\lambda=\lambda_\infty}$, i.e. approximate all vertex functions of order larger then $m_c$ by their initial condition. This generates a closed system of ODEs for $\Gamma^{(0)}, \dots \Gamma^{(m_c)}$ that is exact up to $\mathcal{O}(k^{m_c})$,[4] but which sums up an entire subset of diagrams. This scheme is also known as FRG enhanced perturbation theory [26]. Here we will consider $m_c = 2, 3$ in the quantum case, which leads to results that are exact up to $\mathcal{O}(k^2)$ and $\mathcal{O}(k^3)$ respectively and $m_c = 2, \dots, 5$ in the classical case.

---

[4]Here $k$ is the strength of the interaction in which the truncation occurs. This can be shown employing a diagrammatic argument [26, 32].

Using the shorthand $G_\lambda = G_\lambda^{(2),c}$ and defining the single scale propagator $S_\lambda = G_\lambda Q_\lambda G_\lambda$ the flow equations up to $m_c = 3$ read

$$\dot{\Gamma}^{(0)} = \frac{1}{2}\mathrm{Tr}\left[Q_\lambda(G_\lambda - \mathcal{G}_0^\lambda)\right] - \left[G_0^\lambda(0)\right]^{-1}\phi_\lambda^\star\left(\partial_\lambda\phi_\lambda^\star\right), \tag{20}$$

$$\left(\partial_\lambda\phi_\lambda^\star\right)G_\lambda(0)^{-1} = \frac{1}{2}\mathrm{Tr}\left[S_\lambda\Gamma_{\cdot,0,\cdot}^{(3)}\right] - \phi_\lambda^\star Q_\lambda(0), \tag{21}$$

$$\dot{\Gamma}_{i,-i}^{(2)} - \left(\partial_\lambda\phi_\lambda^\star\right)\Gamma_{0,i,-i}^{(3)} = -\frac{1}{2}\mathrm{Tr}\left[S_\lambda\Gamma_{\cdot,i,-i,\cdot}^{(4)}\right] + \mathrm{Tr}\left[S_\lambda\Gamma_{\cdot,i,\cdot}^{(3)}G_\lambda\Gamma_{\cdot,-i,\cdot}^{(3)}\right], \tag{22}$$

$$\dot{\Gamma}_{i,j,\nu}^{(3)} - \left(\partial_\lambda\phi_\lambda^\star\right)\Gamma_{0,i,j,\nu}^{(4)} = -\frac{1}{2}\mathrm{Tr}\left[S_\lambda\Gamma_{\cdot,i,j,\nu,\cdot}^{(5)}\right] \tag{23}$$

$$+ \mathrm{Tr}\left[S_\lambda\Gamma_{\cdot,i,j,\cdot}^{(4)}G_\lambda\Gamma_{\cdot,\nu,\cdot}^{(3)}\right] + \mathrm{Tr}\left[(i \leftrightarrow \nu)\right] + \mathrm{Tr}\left[(j \leftrightarrow \nu)\right]$$

$$- \frac{1}{2}\left\{\mathrm{Tr}\left[S_\lambda\Gamma_{\cdot,i,\cdot}^{(3)}G_\lambda\Gamma_{\cdot,j,\cdot}^{(3)}G_\lambda\Gamma_{\cdot,\nu,\cdot}^{(3)}\right]\right\} + \left\{\mathrm{Perm.} \in S(\{i,j,\nu\})\setminus 1\right\},$$

$$\lim_{\lambda\to\lambda_\infty}\Gamma_{i_1,\dots,i_n}^{(n)} = \left\{\frac{\delta}{\delta\phi(i_1)}\cdots\frac{\delta}{\delta\phi(i_n)}S_{\mathrm{int}}(\phi)\right\}\Bigg|_{\phi=\phi_{\lambda_\infty}^\star}, \tag{24}$$

where in the last equation $\nu = -i - j$, because each vertex-function imposes frequency conservation. The Tr is to be taken over the Matsubara-frequencies. The initial conditions at $\lambda = \lambda_\infty$ are obtained by inserting the Taylor-expansion (19) into (18) and comparing coefficients. We have rewritten the second equation for $\Gamma^{(1)}$ in terms of $\phi_\lambda^\star$ by using the equation of state from we also directly obtain the initial condition for $\phi_\lambda^\star$

$$\Gamma^{(1)} \equiv \frac{\delta\Gamma^\lambda}{\delta\phi}\Bigg|_{\phi=\phi^\star} = -\left[\mathcal{G}_0^\lambda\right]^{-1}\phi_\lambda^\star \implies \lim_{\lambda\to\lambda_\infty}\mathcal{G}_0^\lambda S_{\mathrm{int}}^{(1)}(\phi^\star) = 0 = -\phi_{\lambda_\infty}^\star. \tag{25}$$

A common choice for the regulator at $T = 0$ is $f_\lambda(\omega) = \theta(|\omega| - \lambda)$, which in combination with the Morris-lemma [33] leads to a particularly convenient form of the flow equations (if $\phi_\lambda^\star = 0$) since there are no remaining frequency-integrals on the right hand side. Here we have $\phi_\lambda^\star \neq 0$, which generates additional terms in the flow equations. These additional terms (e.g. $\phi_\lambda^\star G_\lambda(0)Q_\lambda(0)$) diverge when trying to apply the Morris Lemma for a sharp cutoff.[5] A solution for this issue is to use a soft cutoff, which leads to numerically more expensive equations, since one retains integrals on the right hand side, but circumvents the divergences.

An alternative to a soft cutoff was presented in [24], where the authors redefine the free propagator, such that the problematic terms in the flow equations vanish identically.[6] For our truncation schema such a redefinition is insufficient, as it amounts to grouping the $\sim x^2$ term into the interaction. This would in turn mean that the FRG sums up diagrams with the free particle $H_0 = p^2/2$ as the non-interacting problem (no harmonic oscillator potential). An explicit form of the flow equations and details of the numerical implementation to solve them can be found in appendix A.

The classical integral does not have any internal structure (no Matsubara-frequencies), making it feasible to include higher orders in the truncation. Below we show the flow equations up to $m_c = 5$

$$\dot{\Gamma}^{(0)} = \frac{1}{2}Q_\lambda\left(G_\lambda - G_0^\lambda\right) - \left(\partial_\lambda\phi_\lambda^\star\right)\phi_\lambda^\star\left[G_0^\lambda\right]^{-1}, \tag{26}$$

$$\left(\partial_\lambda\phi_\lambda^\star\right) = \frac{1}{2}Q_\lambda G_\lambda^3\Gamma^{(3)} - Q_\lambda G_\lambda\phi_\lambda^\star, \tag{27}$$

---

[5]The divergence is visible in the Morris-Lemma, as well as when starting with a soft Heaviside function $\theta \to \theta_\epsilon$ and then taking the limit $\epsilon \to 0$ in the end of the derivation.

[6]The derivation of the flow equations in [24] differs slightly from our own, but if we define our free propagator in the way done there, viz. $\mathcal{G}_0^{-1}\phi^\star = 0$ and $Q_\lambda\phi^\star = 0$, our flow equations reduce to the ones reported there.

$$\dot{\Gamma}^{(2)} - \left(\partial_\lambda \phi_\lambda^\star\right)\Gamma^{(3)} = -\frac{1}{2}Q_\lambda G_\lambda^2 \Gamma^{(4)} + Q_\lambda G_\lambda^3 \left(\Gamma^{(3)}\right)^2, \tag{28}$$

$$\dot{\Gamma}^{(3)} - \left(\partial_\lambda \phi_\lambda^\star\right)\Gamma^{(4)} = -\frac{1}{2}Q_\lambda G_\lambda^2 \Gamma^{(5)} + 3Q_\lambda G_\lambda^3 \Gamma^{(4)}\Gamma^{(3)} - 3Q_\lambda G_\lambda^4 \left(\Gamma^{(3)}\right)^3, \tag{29}$$

$$\dot{\Gamma}^{(4)} - \left(\partial_\lambda \phi_\lambda^\star\right)\Gamma^{(5)} = -\frac{1}{2}Q_\lambda G_\lambda^2 \Gamma^{(6)} + 4Q_\lambda G_\lambda^3 \Gamma^{(5)}\Gamma^{(3)} - 18Q_\lambda G_\lambda^4 \Gamma^{(4)}\left(\Gamma^{(3)}\right)^2 \tag{30}$$

$$+ 3Q_\lambda G_\lambda^3 \left(\Gamma^{(4)}\right)^2 + 12Q_\lambda G_\lambda^5 \left(\Gamma^{(3)}\right)^4,$$

$$\dot{\Gamma}^{(5)} - \left(\partial_\lambda \phi_\lambda^\star\right)\Gamma^{(6)} = -\frac{1}{2}Q_\lambda G_\lambda^2 \Gamma^{(7)} + 5Q_\lambda G_\lambda^3 \Gamma^{(6)}\Gamma^{(3)} + 10Q_\lambda G_\lambda^3 \Gamma^{(5)}\Gamma^{(4)} \tag{31}$$

$$- 30Q_\lambda G_\lambda^4 \Gamma^{(5)}\left(\Gamma^{(3)}\right)^2 - 45Q_\lambda G_\lambda^4 \Gamma^{(3)}\left(\Gamma^{(4)}\right)^2$$

$$+ 120Q_\lambda G_\lambda^5 \Gamma^{(4)}\left(\Gamma^{(3)}\right)^3 - 60Q_\lambda G_\lambda^6 \left(\Gamma^{(3)}\right)^5,$$

$$\lim_{\lambda \to \lambda_\infty} \Gamma^{(n)} = \frac{\delta^n}{\delta \phi^n}S_{\text{int}}(\phi)|_{\phi=\phi_{\lambda_\infty}^\star}. \tag{32}$$

We emphasize that the modifications necessary in the FRG-VE are not due to the nH or $\mathcal{PT}$-symmetric nature of the model, but are rather due to $\phi^\star \neq 0$ which also can be present in Hermitian systems, e.g. when studying superfluidity [24].

### 3.2 Other Methods

We want to compare the FRG results with an exact solution. While in the classical case, the model becomes analytically solvable (see section 4), we use exact diagonalization (ED) for the quantum case. We represent the Hamiltonian in the occupation number basis $\{|n\rangle\}$ of the harmonic oscillator and diagonalize the upper-left corner of the matrix $\langle n|H|m\rangle$ with $n, m \leq n_{\text{c}}$. We always made sure that $n_{\text{c}}$ was chosen such that the results are convergent on the scale of the corresponding plots.

Using equation (6) (resolution of unity) the Lehmann-representation for the connected 2-point Greens function at $T = 0$ reads

$$G_\omega^{(2),\text{c}} = \sum_{l>0} \frac{2(E_l - E_0)}{\omega^2 + (E_l - E_0)^2} \langle L_0|x|R_l\rangle \langle L_l|x|R_0\rangle \tag{33}$$

and allows to calculate $G_\omega^{(2),\text{c}}$ if $E_l$, $|R_l\rangle$ and $|L_l\rangle$ are known.

In addition to the exact solution we also compare with perturbation theory which requires no additional modifications for the (connected) Greens functions and similar quantities. When calculating the vertex functions $\Gamma^{(n)}$ perturbativly care has to be taken though, because $\phi^\star \neq 0$, which necessitates the usage of a modified propagator in the diagrams: The starting point here is a honest loopwise-expansion where the expansion point $\phi$ is unconstrained in the beginning [32,34]. The perturbative expansion is then given by evaluating the diagrams of the loopwise-expansion at $\phi = \phi^\star$ and sorting the contributions order by order in the interaction. $\phi^\star$ is determined perturbativly by evaluating all connected diagrams with one external leg. We can thus write the perturbative expansion of the vertex-functions as (suppressing the frequency dependence)

$$\Gamma^{(n)}[\phi] = \Gamma_{\text{fl}}^{(n)}[\phi] + \frac{\delta^n}{\delta \phi^n}\left\{S[\phi] - \frac{1}{2}\left(\phi, \left[\mathcal{G}_0\right]^{-1}\phi\right)\right\}, \tag{34}$$

$$\Gamma_{\text{fl}}^{(n)}[\phi] = -\sum_{\substack{\text{1-PI} \\ n \text{ amp. legs}}} \in \text{graphs}(\Delta, R), \tag{35}$$

$$\Delta^{-1} = \frac{\delta^2 S}{\delta \phi^2}\bigg|_{\phi=\phi^\star}, \quad R = \left\{ n > 2 \,\middle|\, \text{Vertex defined by } \frac{\delta^n S}{\delta \phi^n}\bigg|_{\phi=\phi^\star} \right\}.$$

Finally we want to compare the FRG with mean-field theory. In the quantum case we search for a mean-field (mf) Hamiltonian of the form $H_{\text{mf}} = (\frac{1}{2} + \alpha_{\text{p}})p^2 + (\frac{1}{2} + \alpha_{\text{x}})x^2 + \alpha_{\text{s}}x$, where we added the $\alpha_{\text{s}}$ term to allow for a theory with $\langle x \rangle \neq 0$. The coefficients $\alpha_{\text{m}}$ are determined in Bugoliobov mean-field theory, where one can show that

$$\alpha_{\text{m}} = \frac{\partial \langle S_{\text{int}} \rangle_{\text{mf}}}{\partial \langle A_{\text{m}} \rangle}, \quad \text{with} \quad \langle A_{\text{m}} \rangle = p^2, x^2, x. \tag{36}$$

In a first step we calculate the expectation value of the non-linearity

$$\langle S_{\text{int}} \rangle_{\text{mf}} = \frac{ik}{3!} \langle \big( \underbrace{x - x^\star}_{=:y} + x^\star \big)^3 \rangle_{\text{mf}} = \frac{ik}{3!} \Big( 3x^\star \langle (x - x^\star)^2 \rangle_{\text{mf}} + (x^\star)^3 \Big) \tag{37}$$

$$= \frac{ik}{3!} \Big( 3x^\star \langle x^2 \rangle_{\text{mf}} - 2(x^\star)^3 \Big), \tag{38}$$

where we have shifted the integration variable by its mean-value $\langle x \rangle_{\text{mf}} = x^\star$ at the second equal sign, because the mf-action is Gaussian with $\mu = 0$ in the variable $y = x - x^\star$, allowing the usage of the standard Wick-theorem. Using (36) the mf Hamiltonian reads

$$H_{\text{mf}} = \frac{p^2}{2} + \frac{1}{2}(1 + ikx^\star)\, x^2 + \frac{ik}{2}\big( \langle x^2 \rangle - 2(x^\star)^2 \big)\, x, \tag{39}$$

where the mean fields $x^\star, \langle x^2 \rangle$ are determined by the self-consistency equations at $T = 0$

$$x^\star = -\frac{ik}{2}\big( \langle x^2 \rangle - 2(x^\star)^2 \big), \tag{40}$$

$$\langle x^2 \rangle = \frac{1}{2\sqrt{1 + ikx^\star}} + (x^\star)^2. \tag{41}$$

In the classical case the self-consistency equations are different. The mf action reads (again only taking the $\omega = 0$ component of the quantum action)

$$S_{\text{mf}} = \frac{1}{2}\big(1 + ikx^\star\big)x^2 + \frac{ik}{2}\big( \langle x^2 \rangle - 2(x^\star)^2 \big), \tag{42}$$

and the mean fields are determined by

$$x^\star = \frac{1}{Z_{\text{mf}}} \int dx\, x\, e^{-S_{\text{mf}}(x)} = -\frac{ik}{2} \frac{\langle x^2 \rangle - 2(x^\star)^2}{1 + ikx^\star}, \tag{43}$$

$$\langle x^2 \rangle_{\text{mf}} = \frac{1}{Z_{\text{mf}}} \int dx\, x^2\, e^{-S_{\text{mf}}(x)} = \frac{1}{1 + ikx^\star} + (x^\star)^2. \tag{44}$$

We solve the self-consistency equations in both the quantum and the classical case with a standard fixed point iteration.

### 3.3 Observables

Solving the FRG flow equations we obtain the vertex functions. While in the classical case we will directly compare the vertex functions with their exact solution, in the quantum case we

are primarily interested to compare eigen- or expectation values of physical 'observables'. One can show [26] that

$$e_0 = E_0 - E_0|_{k=0} = \Gamma^{(0)} + \frac{1}{2}\left(\beta^{-\frac{1}{2}}\phi^\star\right)^2,\tag{45}$$

$$\beta^{-\frac{1}{2}}\phi^\star = \langle x(\tau = 0)\rangle,\tag{46}$$

$$\langle\langle x^2\rangle\rangle = \langle x^2\rangle - \langle x\rangle^2 = \int \frac{d\omega}{2\pi}\frac{1}{\mathcal{G}_0^{-1}(\omega) + \Gamma_\omega^{(2)}}.\tag{47}$$

Furthermore we will compare the first excitation energy $e_1 = E_1 - E_1|_{k=0}$ which we obtain from the Greens function as follows (see [26] for details): We approximate the Greens function by the first term in the Lehmann representation (33), i.e. $G_\omega^{(2),c} \approx \frac{a}{\omega^2+b}$. This is a valid approximation, because the Lehmann representation in (33) is dominated by the first term ($l = 1$), the next term in the sum ($l = 2$) already being two orders of magnitude smaller.[7]

Fitting this form to the FRG Greens function yields $e_1$. The validity of this procedure can be assessed by comparing $e_1$ calculated directly from ED and $e_1$ extracted by this fitting procedure applied to the ED Greens function. We find a fairly good agreement, see figure 3 top-right.

Following [27] we define an operator $A = \sum_{\alpha\beta} A_{\alpha\beta}|R_\alpha\rangle\langle L_\beta|$ to be an observable, if and only if $A_{\alpha,\beta}$ is Hermitian. This guarantees that the expectation value of $A$ (in the sense of biorthogonal quantum mechanics [27]) in an arbitrary state is real. While according to this definition $H$ is obviously a physical observable ($H_{\alpha,\beta}$ being diagonal) and $e_0$, $e_1$ are directly related to the eigenvalues, $x$ and $\langle\langle x^2\rangle\rangle = \left(x - \langle x\rangle\right)^2$ do not fall into this class of operators ($\langle x\rangle$ is purely imaginary). The expectation values of the latter two operators can nevertheless be used to judge the quality of the approximation. In addition, they carry information about symmetry breaking [30] in models in which such occurs (not the case for our model. See section 5).

# 4 Results

## 4.1 Classical Integral

The exact reference for the 'classical' integral is an analytical solution of (11). Defining $I_n = (2\pi)^{-1}\int_{-\infty}^{\infty} dx\, x^n e^{-S(x)}$ we find for $k \in \mathbb{R}^+$ with $K_n(z)$ being the modified Bessel-function of 2nd kind

$$I_0(k) = \sqrt{\frac{2}{3\pi}}\frac{e^{\frac{1}{3k^2}}}{k}K_{\frac{1}{3}}\left(\frac{1}{3k^2}\right),\tag{48}$$

$$I_1(k) = i\sqrt{\frac{2}{3\pi}}\frac{e^{\frac{1}{3k^2}}}{k^2}\left\{K_{\frac{1}{3}}\left(\frac{1}{3k^2}\right) - K_{\frac{2}{3}}\left(\frac{1}{3k^2}\right)\right\},\tag{49}$$

$$I_2(k) = -2\sqrt{\frac{2}{3\pi}}\frac{e^{\frac{1}{3k^2}}}{k^3}\left\{K_{\frac{1}{3}}\left(\frac{1}{3k^2}\right) - K_{\frac{2}{3}}\left(\frac{1}{3k^2}\right)\right\},\tag{50}$$

$$I_3(k) = -2i\sqrt{\frac{2}{3\pi}}\frac{e^{\frac{1}{3k^2}}}{k^4}\left\{(2+k^2)K_{\frac{1}{3}}\left(\frac{1}{3k^2}\right) - 2K_{-\frac{2}{3}}\left(\frac{1}{3k^2}\right)\right\},\tag{51}$$

$$I_4(k) = 4\sqrt{\frac{2}{3\pi}}\frac{e^{\frac{1}{3k^2}}}{k^5}\left\{2(1+k^2)K_{\frac{1}{3}}\left(\frac{1}{3k^2}\right) - (2+k^2)K_{\frac{2}{3}}\left(\frac{1}{3k^2}\right)\right\}.\tag{52}$$

---

[7]This implies that the contributions of higher-order excitations to $G_\omega^{(2),c}$ and $\Gamma_\omega^{(2)}$ are very small such that an analytic continuation $i\omega_n \to \omega + i0^+$ of the FRG results, with e.g. an Padé approximation, will not allow an accurate approximation of the full spectrum.

These integrals represent the usual Greens functions for the toy-theory defined by $G_m = \frac{I_m}{I_0}$. They can thus be related to the connected Greens functions $G_m^c = \frac{\partial^m W(j)}{\partial j^m}$ and hence to the vertex functions $\Gamma^{(m)} = \frac{\partial^m \Gamma}{\partial \phi^m}$ in the standard way [32].

For the classical integral we want to compare the vertex functions calculated with the different approximations directly to the exact result. The relevant relations read

$$\Gamma^{(0)} = -\log(I_0) - \frac{1}{2}G_1^2\,, \tag{53}$$

$$\Gamma^{(1)} = -G_1 = -\phi^\star\,, \tag{54}$$

$$\Gamma^{(2)} = \frac{1}{G_2 - G_1^2} - 1\,, \tag{55}$$

$$\Gamma^{(3)} = -\frac{1}{\left(1 + \Gamma^{(2)}\right)^3}\left(G_3 - 3G_2 G_1 + 2G_1^3\right)\,, \tag{56}$$

$$\Gamma^{(4)} = -\frac{1}{\left(1 + \Gamma^{(2)}\right)^4}\left(G_4 - 4G_3 G_1^2 - 3G_2^2 + 12G_2 G_1^2 - 6G_1^4\right) + 3\frac{\left(\Gamma^{(3)}\right)^2}{1 + \Gamma^{(2)}}\,. \tag{57}$$

The perturbative expansion of the vertex functions, obtained form diagrammatic perturbation theory read

$$\Gamma^{(0)} = \frac{k^2}{3} - \frac{5}{8}k^4 + \mathcal{O}\left(k^6\right)\,, \tag{58}$$

$$\phi^\star = -\frac{ik}{2} + \frac{5}{8}ik^3 - \frac{15}{8}ik^5 + \mathcal{O}\left(k^6\right)\,, \tag{59}$$

$$\Gamma^{(2)} = k^2 - \frac{17}{8}k^4 + \mathcal{O}\left(k^6\right)\,, \tag{60}$$

$$\Gamma^{(3)} = ik - ik^3 + \frac{13}{2}ik^5 + \mathcal{O}\left(k^6\right)\,, \tag{61}$$

$$\Gamma^{(4)} = -3k^4 + 36k^6 + \mathcal{O}\left(k^8\right)\,. \tag{62}$$

These results have also been used as a consistency check for the FRG implementation: Since the flow equations truncated at $m_c$ are exact up to $\mathcal{O}(k^{m_c})$, one can reproduce the Taylor-coefficients up to $\mathcal{O}(k^{m_c})$ from the numerical solution of the FRG equations (see Appendix B).

For the FRG we use an exponential cutoff of the form $f_\lambda = e^{-\lambda}$ and integrate the system from $\lambda_\infty = 30$ to $\lambda_0 = 0$ where we made sure the results are converge in $\lambda_\infty$. We also used a linear cutoff $f_\lambda = \lambda$ with $\lambda_0 = 10^{-10}, \lambda_\infty = 1$ and found the results to be equivalent on the scale of the plots.

In addition to applying the FRG equations derived in section 3 we also performed a vertex expansion around $\phi^\star = 0$, that is with a fixed expansion point.[8] The resulting solution does not even reproduce the Taylor-coefficients of order less then $m_c$, so one must indeed consider a flowing expansion point if $\phi^\star \neq 0$.

The results of our analysis are summarized in figure 2. Whilst perturbation theory already fails for $k \sim 1$, the other approximation schemes are significantly more accurate up to large couplings $k \sim 20$. FRG truncated at $m_c = 2$ and mf yield comparable results (FRG being slightly more accurate). The higher orders $m_c = 3, \ldots, 5$ are significantly better approximations then mf. Especially $m_c = 5$ only shows very minor deviations from the exact result. All this holds despite the fact that also in the truncated FRG only the first $m_c$ derivatives at $k = 0$

---

[8] Note that $\phi^\star \neq 0$, e.g. determined by perturbation theory, is not a valid expansion point since at the beginning of the flow $\phi^\star_{\lambda_\infty} = 0$. See equation (25).

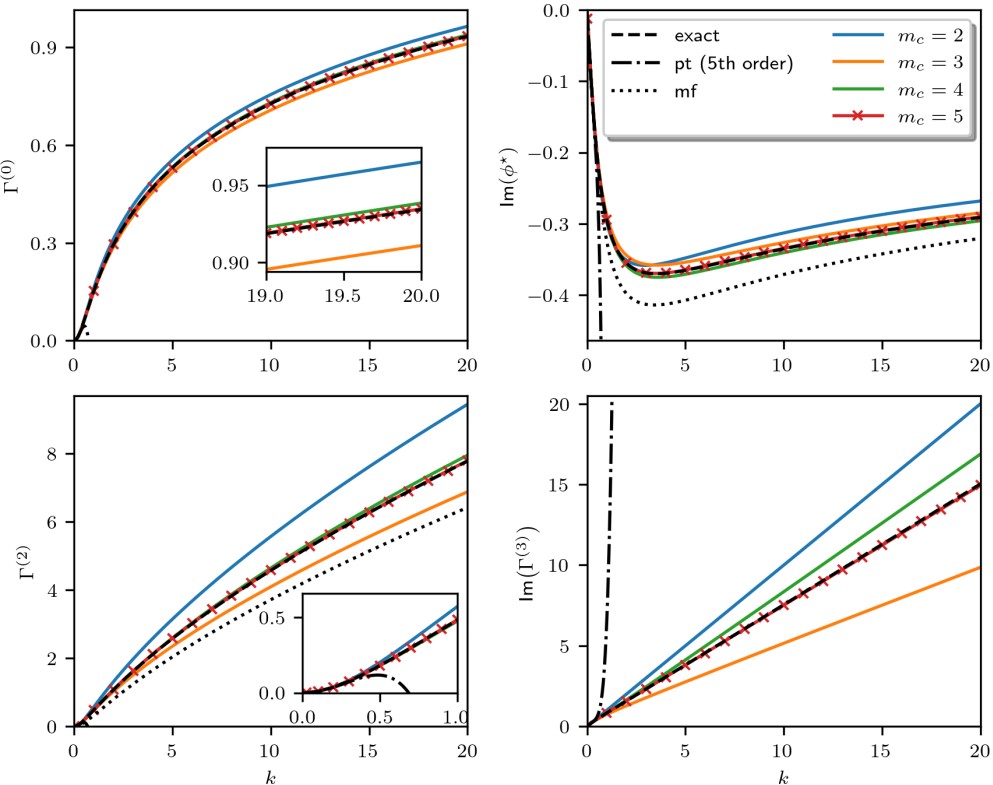

Figure 2: Results for classical integral. Exact solution (dashed line) obtained from an analytical calculation of the integral are compared with perturbation theory (pt, dash-dotted), mean-field theory (mf, dotted) and the FRG (full, color-coded) truncated at $m_c = 2, \dots 5$.

are reproduced correctly. This provides a first indication that the FRG can indeed be applied for $\mathcal{PT}$-symmetric systems, yielding results comparable in quality to the Hermitian case [20].

## 4.2 Quantum Theory

For an exact solution of the quantum theory we use exact diagonalization as outlined in section 3. We find all results to be converged on the scale of the plots (see below) for a truncation of the Hilbert-space at $n_c = 1000$.

The perturbative results obtained from a diagrammatic expansion read[9]

$$e_0 = E_0 - E_0|_{k=0} = \frac{11}{288}k^2 - \frac{155}{13824}k^4 + \mathcal{O}(k^6), \tag{63}$$

$$\langle x \rangle = -\frac{ik}{4} + \frac{11}{144}ik^3 + \mathcal{O}(k^5), \tag{64}$$

$$\Gamma^{(2)}_{\omega,-\omega} = \left(\frac{1}{4} + \frac{1}{2}\frac{1}{4+\omega^2}\right)k^2 + \frac{3420 + 1259\omega^2 + 200\omega^4 - 11\omega^6}{144(4+\omega^2)^2(9+\omega^2)}k^4 + \mathcal{O}(k^6), \tag{65}$$

$$\langle\langle x^2 \rangle\rangle = \frac{1}{2} - \frac{13}{144}k^2 + \frac{289}{4608}k^4 + \mathcal{O}(k^6), \tag{66}$$

$$\Gamma^{(3)}_{\omega,\nu,\delta} = ik - ik^3\frac{12 + \omega^2 + \nu^2 + \nu\omega}{(4+\omega^2)(4+\nu^2)(4+(\omega+\nu)^2)} + \mathcal{O}(k^5). \tag{67}$$

---

[9]The results for $e_0$ and $\langle x \rangle$ have also been obtained in [22].

These results have also been used as a consistency check for the ED and the FRG as explained in 4.1 (see Appendix B).

For the FRG we use a linear cutoff $f_\lambda(\omega) = \lambda$ and integrate the system from $\lambda_\infty = 10^{-10}$ to $\lambda_0 = 1$. We made sure that the results are converged for the chosen $\lambda_\infty$. Our guiding principle for the cutoff choice (besides fidelity of the results) is numerical stability. We tested multiple cutoffs, including frequency dependent ones (for $m_c = 2$) but found that they do not influence the fidelity of the results on the scale of the plots.[10] But the cutoff does influence the number of steps the integrator requires at given accuracy, as well as the stability of the ODE system. We obtained the best integration time and stability with the linear cutoff. There are systematic ways for choosing an optimized cutoff, such as the principle of minimal sensitivity [35], but since we find no cutoff dependence on the scale of the plots, we will not pursue this any further here.

The behavior of the vertex-functions along the flow, i.e. as a function of $\lambda$, is shown in figure 1 for $k = 3$. Our results, comparing the observables at different $k$, are summarized in figure 3.

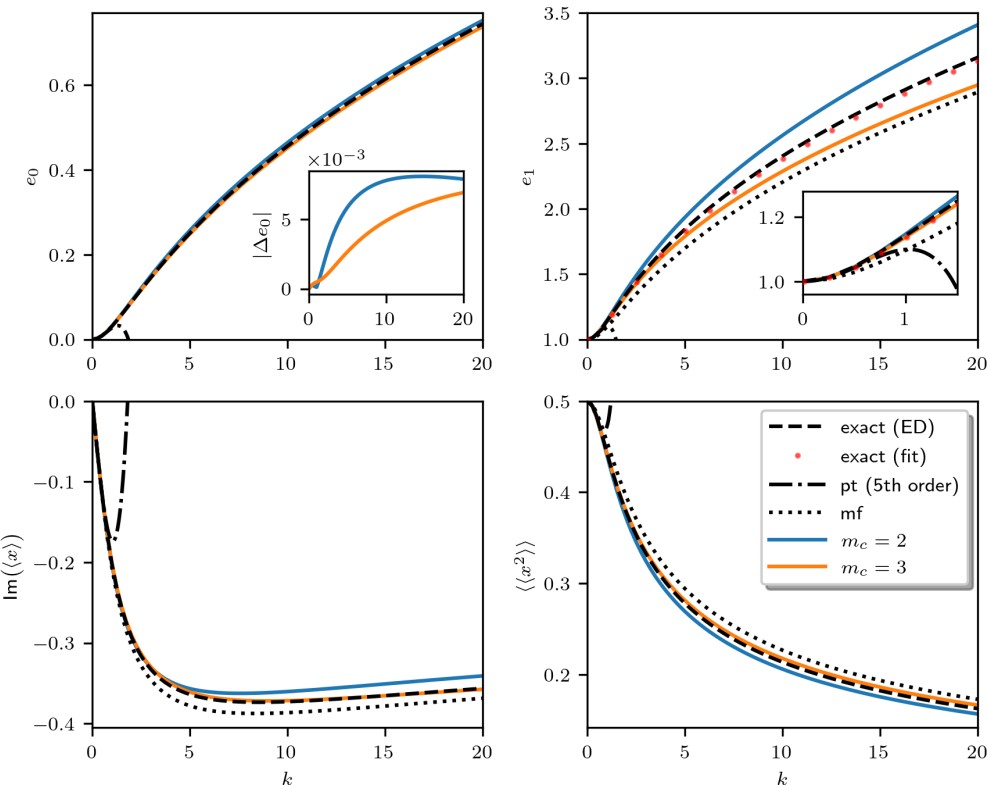

Figure 3: Results for quantum theory. We show the ground state-energy $e_0 = E_0 - E_0|_{k=0}$, first excitation energy $e_1 = E_1 - E_1|_{k=0}$, vacuum expectation value $\langle x \rangle$ and fluctuations $\langle\langle x^2 \rangle\rangle$. Exact results obtained from ED are compared with perturbation theory (pt, dash-dotted), mean-field theory (mf, dotted) and FRG (full, color-coded) truncated at $m_c = 2, 3$. Exact fit denotes the calculation of $e_1$ from the exact propagator as outlined in section 3. The inset in the top-left shows the absolute-difference between the exact $e_0$ and FRG.

---

[10] We used several cutoffs, e.g. of exponential type $f_\lambda(\omega) = \exp\left(-\lambda\left[1 + \frac{1}{1+0.1\cdot\omega^2}\right]\right)$ or of Heaviside type $f_\lambda(\omega) = \left(\exp\left(\frac{\lambda}{1+0.1\cdot\omega^2}\right) + 1\right)^{-1}$.

The red-dotted line in the top-right graph of 3 shows $e_1$ obtained from the propagator determined within ED with the fitting method outlined in section 3. The agreement with the black-dashed line, which shows the $e_1$ directly obtained from ED, is not perfect, but good enough to use this method as a means to obtain $e_1$ from the FRG. Taking more terms of the Lehmann representation into account leads to instabilities in the parameter fitting, so that we refrain from doing so here.

As in the classical case, perturbation theory (pt) is only reliable for small couplings $k \sim 1$ whilst the other schemes produce reasonable approximations even in the strong coupling regime $k \sim 20$. The results for mf and $m_c = 2$ are again comparable, the FRG being slightly better. One possible reason for this improvement could be that the self-energy $\Sigma = -\Gamma^{(2)}$ in this truncation order includes a frequency dependency, while $\Sigma_{\mathrm{mf}}$ does not. Truncation at $m_c = 3$ gives an excellent approximation to the exact results for all quantities excepts $e_1$. But even there it still provides the best results from the considered methods. As in the classical case this shows, that FRG can also be applied to $\mathcal{PT}$-symmetric systems, where one can expect a similar fidelity of the approximation as in the Hermitian case (see [26] for a study of the anharmonic oscillator).

## 5 Conclusion

The goal of this work was to generalize the vertex expansion (VE) of the functional renormalization group (FRG) to non-Hermitian (nH) systems. Assuming the biorthogonal eigenbasis of the Hamiltonian to be complete we showed that the FRG-VE flow equations can be derived for a general nH system. The presence of a non-vanishing vacuum expectation value $\langle x \rangle \neq 0$ in our model, lead to some modifications of the flow-equations. As a first test of the FRG-VE for nH systems we analyzed a $\mathcal{PT}$-symmetric quantum system with a non-linearity in the class introduced in [3] that is free of $\mathcal{PT}$-symmetry breaking and which is exactly solvable, thus providing us with a controlled testbed for a first applications of the FRG-VE. Here the non-linearity of the studied model can be viewed as a proxy for interactions/correlations in a many-body context.

We compared the exact solution of the model, obtained via an exact diagonalization, with mean-field theory, perturbation theory and the FRG-VE truncated at different orders. We find the FRG-VE, truncated at the 3-point vertex, to be the best approximation considered here, even up to large couplings. Further the fidelity of the FRG-VE is comparable to the Hermitian case so that it seems to present a viable option for studying correlation effects in nH systems.

Previous studies of correlated nH systems have also found interesting new physics at the exceptional point and in the phase of broken $\mathcal{PT}$-symmetry [5,10–13]. Further, in a correlated system it might not be obvious if the system shows $\mathcal{PT}$-symmetry breaking, such that an important extension of our work will be to analyze if the FRG-VE can also capture $\mathcal{PT}$-symmetry breaking and if it can be applied in the phase of broken $\mathcal{PT}$-symmetry. A prime system for studying this in the same schema introduced in this paper was introduced in [36], where the authors analyze $\mathcal{PT}$-symmetry breaking in systems of 2 and 3 coupled oscillators using ED.

Furthermore, our development allows to analyze correlation effects in real many-body systems and to study transport-phenomena. An interesting candidate for this would be the interacting resonant level model (IRLM) with a $\mathcal{PT}$-symmetric interaction [9]. Further it will be interesting to investigate explicitly time dependent nH systems, e.g. in the context of periodic driving or quenches [37]. Here one can also expect profound modifications of physics due to interactions, whose influence on time dependent transport have been successfully described within the FRG-VE in the Hermitian case [38]. In conclusion, there are many open questions associated with correlations in nH systems and the FRG-VE appears to be a viable tool for studying them.

# Acknowledgements

**Funding information** This work was supported by the Deutsche Forschungsgemeinschaft (DFG, German Research Foundation) via RTG 1995 and Germany's Excellence Strategy - Cluster of Excellence Matter and Light for Quantum Computing (ML4Q) EXC 2004/1 - 390534769. Simulations were performed with computing resources granted by RWTH Aachen University under project rwth0710. We acknowledge support from the Max Planck-New York City Center for Non-Equilibrium Quantum Phenomena.

# A Explicit Flow Equations and Implementation Details

Here we present the explicit form of the flow equations (20) - (24) in the quantum case and discuss the details of the numerical implementation for which we use Julia [39]. We introduce $\gamma^{(3)} = \beta^{-\frac{1}{2}}\Gamma^{(3)}$ and $\gamma^{(4)} = \beta^{-1}\Gamma^{(4)}$ and consider truncations up to $m_c = 3$, meaning we neglect the flow of vertex functions with $n > 3$ by setting their value to the inital-value of the flow; here $\gamma^{(n>3),\lambda} \to \gamma^{(n>3),\lambda_\infty} = 0$. The flow equations for a general cutoff function $f_\lambda(\omega)$ with free propagator $\mathcal{G}_0^{-1}(\omega) = \omega^2 + 1$ then read (shorthand $\int_\omega = \int_{-\infty}^{\infty} \frac{d\omega}{2\pi}$)

$$\beta^{-1}\dot{\Gamma}^{(0)} = \frac{1}{2}\int_\omega f_\lambda'(\omega)\frac{\Gamma_\omega^{(2)}}{\omega^2 + 1 + f_\lambda(\omega)\Gamma_\omega^{(2)}} - \left(\beta^{-\frac{1}{2}}\phi_\lambda^\star\right)\frac{1}{f_\lambda(0)}\left(\partial_\lambda\beta^{-\frac{1}{2}}\phi_\lambda^\star\right),$$

$$\left(\partial_\lambda\beta^{-\frac{1}{2}}\phi_\lambda^\star\right) = \frac{1}{1 + f_\lambda(0)\Gamma_0^{(2)}}\left(-\frac{1}{2}f_\lambda(0)\int_\omega f_\lambda'(\omega)\frac{\omega^2 + 1}{\left(\omega^2 + 1 + f_\lambda(\omega)\Gamma_\omega^{(2)}\right)^2}\gamma_{0,\omega}^{(3)} + \left(\beta^{-\frac{1}{2}}\phi_\lambda^\star\right)\frac{f_\lambda'(0)}{f_\lambda(0)}\right),$$

$$\dot{\Gamma}_i^{(2)} - \left(\partial_\lambda\beta^{-\frac{1}{2}}\phi_\lambda^\star\right)\gamma_{0,i}^{(3)} = -\int_\omega f_\lambda'(\omega)\frac{\omega^2 + 1}{\left(\omega^2 + 1 + f_\lambda(\omega)\Gamma_\omega^{(2)}\right)^2}\frac{f_\lambda(\omega - i)}{(\omega - i)^2 + 1 + f_\lambda(\omega - i)\Gamma_{\omega-i}^{(2)}}\gamma_{i,-\omega}^{(3)}\gamma_{-i,\omega}^{(3)},$$

$$\dot{\gamma}_{i,j,\nu}^{(3)} = \frac{1}{2}\int_\omega f_\lambda'(\omega)\frac{\omega^2 + 1}{\left(\omega^2 + 1 + f_\lambda(\omega)\Gamma_\omega^{(2)}\right)^2}\frac{f_\lambda(\omega - i)}{(\omega - i)^2 + 1 + f_\lambda(\omega - i)\Gamma_{\omega-i}^{(2)}}\frac{f_\lambda(\omega + \nu)}{(\omega + \nu)^2 + 1 + f_\lambda(\omega + \nu)\Gamma_{\omega+\nu}^{(2)}}$$

$$\times \gamma_{i,-\omega}^{(3)}\gamma_{j,i-\omega}^{(3)}\gamma_{\nu,\omega}^{(3)} + \left\{\text{Perm.} \in S\big(\{i, j, \nu\}\big) \setminus 1\right\},$$

where we have omitted the argument which is fixed by frequency conservation.

For the numerical implementation we use a logarithmic grid around 0 defined by

$$\omega_k = \omega_{\max}\frac{2k - N}{N}\exp\left(\frac{|N - 2k| - N}{S}\right), \quad k = 0, 1, \ldots, N. \tag{68}$$

We find the results to be converged on the scale of the plots for $N = 60, \omega_{\max} = 50, S = 35$. Overall the results are fairly insensitive to the choice of $N, S$, as long as $N > 30$; different choices yielding deviations $\sim 10^{-3}\%$ (checked for $k = 0.4, 20$).

The integrals on the right hand side of the flow equations are calculated with an adaptive Gauss-Kronrod quadrature (implemented in QuadGk.jl), where we evaluate values of the vertex function between grid-points by a 3rd-order spline interpolation (implemented in Dierckx.jl). For values outside of the grid (i.e. $|x| > \omega_{max}$) we use a nearest neighbor interpolation for the vertex functions which allows an analytic evaluation of the boundary-integrals. In practice we calculate these integrals numerically once per function call since the analytical expressions are rather complicated. The system of differential equation is solved using Verner's

Most Efficient 7/6 Runge-Kutta method (implemented in DifferentailEquations.jl [40]). As a sanity check we also use a linearly spaced frequency-grid as well as employing high-order Gaussian-quadratures (order $\sim 10^4$) to evaluate the integrals on the right hand side of the flow equations. Additionally we calculated the integrals via a 3rd order interpolation instead of Gaussian-quadrature. All modifications give the same results as the approach outlined above within numerical precision (or 0.1% relative deviation which we deem negligible).

## B  Consistency Checks for FRG Treatment

As mentioned in the main text, the flow equations truncated at $m_c$ are exact up to $\mathcal{O}(k^{m_c})$. This means in particular that one can reproduce the Taylor-coefficients (given in section 4) from the numerical solution of the FRG equations. To extract the (first non-vanishing) Taylor-coefficient of $\mathcal{O}(k^s)$ for some quantity $\mathcal{Q}$ we plot $\mathcal{Q}/k^s$. For $k \to 0$ the intersection of the curve with $k = 0$ is then the corresponding coefficient as obtained from the numerical data. To obtain a coefficient of higher order, we first subtract the lower order coefficients multiplied by the corresponding power of $k$ (in the plots compactly denoted as $\Delta\mathcal{Q}$) and then repeat the above procedure. This is a very sensitive consistency check for both the flow equations as well as for the implementation. Next to the FRG we also use this method as a consistency check for the correct implementation of the exact solution. The results for the quantum case are shown in figure 4 and the results for the classical case in figure 5.

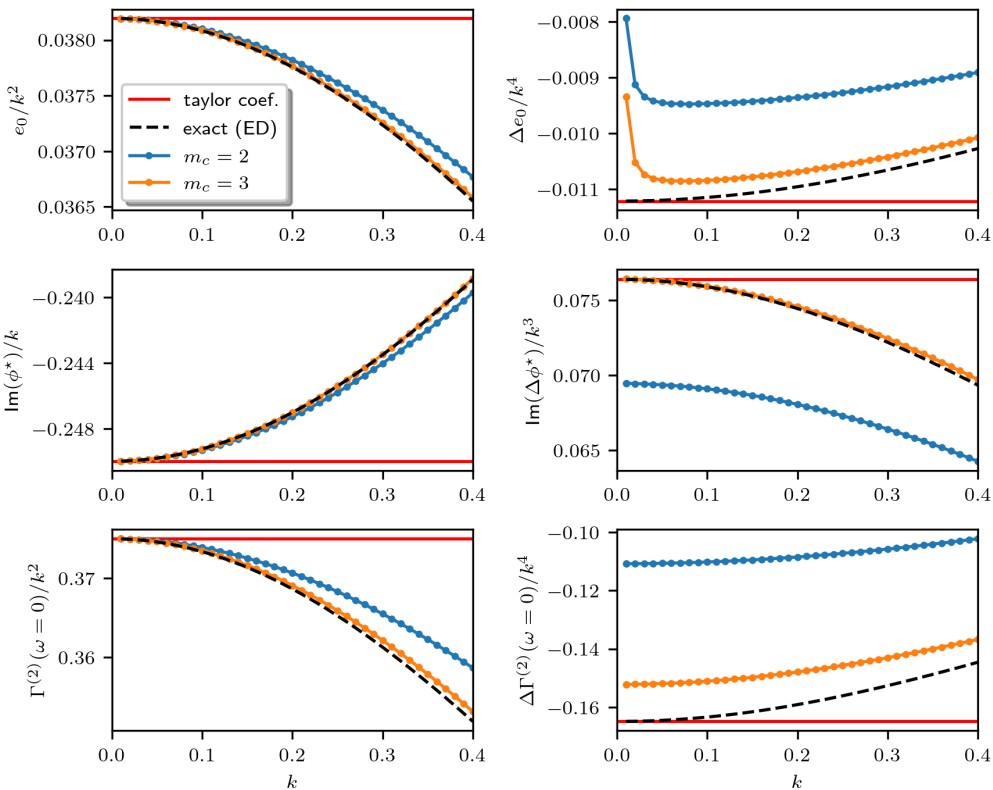

Figure 4: Taylor Coefficients from numerical implementations. We extract them from ED (dashed line) as well as from FRG. We expect the truncation at $m_c$ to correctly reproduce coefficients of order $\mathcal{O}(k^{m_c})$. $\Delta\mathcal{Q}$ denotes the quantity $\mathcal{Q}$ where the lower order Taylor coefficients were subtracted (see main text for a more precise definition).

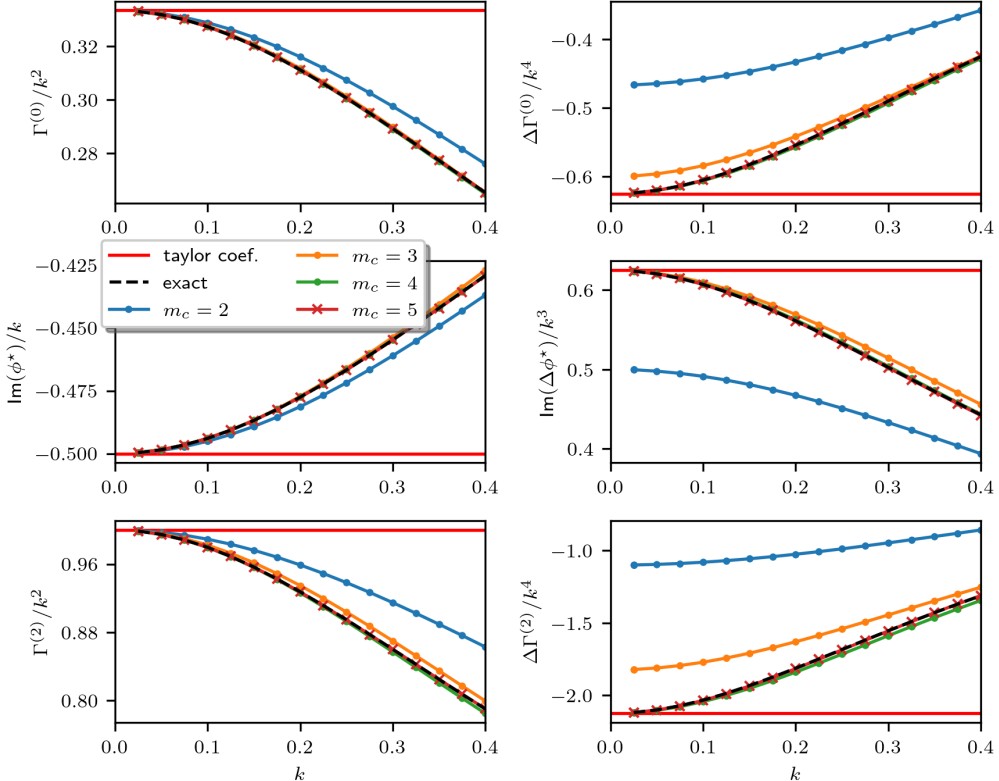

Figure 5: Taylor Coefficients from FRG in classical case. Definition of quantities the same as in figure 4.

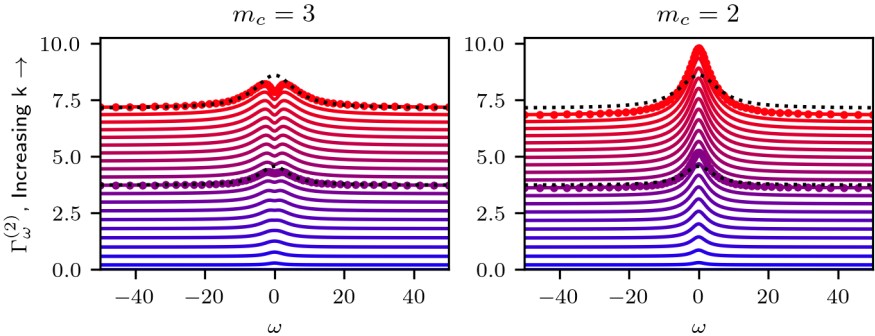

Figure 6: Comparison of $\Gamma^{(2)}_\omega$ calculated with FRG in truncation $m_c = 2, 3$ with exact results obtained from ED. We note the development of a dip in the $m_c = 3$ case. Black dotted lines are ED results for $k = 10$ (lower line) and $k = 20$ (upper line). The FRG results corresponding to these couplings are plotted with additional markers on top of the line.

## C  Comparison of Self-Energies

In figure 6 we compare the frequency dependence of the 2-point vertex function $\Gamma^{(2)}_\omega = -\Sigma_\omega$ with the results obtained from ED. While in the case of $m_c = 3$ a dip develops for small $\omega$ and large $k$ which is not present in the exact solution, the overall fit is significantly better then in the case of $m_c = 2$.

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
