# Peer review of "Functional renormalization group for non-Hermitian and PT-symmetric systems"

_SciPost Physics, doi:SciPost Phys. 12, 179 (2022)_

## Round 1 · Referee Report · Anonymous (Referee 1) · 2022-4-7

Report

The authors generalize the vertex expansion of the functional renormalization group (fRG) to systems with a non-Hermitian Hamiltonian. They analyze the performance of truncated expansions by applying it to an exactly solvable PT-symmetric toy-model, and find that the accuracy is comparable to that for similar Hermitian models.

There is increasing recent interest in non-Hermitian models as effective models, e.g., for open quantum systems. It is thus good to see that the fRG can be applied to such systems, too, at least under suitable circumstances.

The paper seems scientifically sound and is generally well written. I have noticed only two issues which I believe should be clarified before publication.

1) Below Eq. (4) the authors write that they ASSUME that the eigenvalues are not degenerate. Since the model (2) can be solved exactly, it should be possible to clarify whether its eigenvalues exhibit degeneracies or not.

2) I am worried by the divergenies encountered for a sharp cutoff, as mentioned below Eq. (26). Circumventing this problem by choosing a smooth cutoff is not fully satisfactory, since the results will then obviously depend a lot on how smooth the cutoff is, that is, there is a strong dependence on the cutoff function. I suspect that the divergence could be avoided by shifting the expansion point via a suitable counterterm. It is well known that even the perturbation theory for classical anharmonic oscillators fails miserably if one does not shift the oscillator frequency. Choosing an optimal (quadratic) expansion point might also improve the convergence of the expansion. I would like the authors to investigate this point more thoroughly.

In summary, I recommend publication in SciPost Physics after the above-mentioned points have been taken into account.

  • validity: top
  • significance: high
  • originality: high
  • clarity: high
  • formatting: perfect
  • grammar: perfect

Author:  Lukas Grunwald  on 2022-05-05  [id 2442]

(in reply to Report 1 on 2022-04-07)
Category:
remark
answer to question

Dear Referee,
thank you very much for your comments. Regarding the issues you pointed out in your report:

1.) The section starting just above eq. (4) is meant to describe the structure of QM for a general non-hermitian Hamiltonian, so is not specific to our model. We assume non-degenerate eigenvalues only for notational convenience, but for our model there is indeed no degeneracy. We have made this clearer in the new version of the manuscript.

2.) The appearance of the divergence when using a sharp-cutoff is essentially a technical problem. It only appears in the limit of a sharp cutoff, meaning that the flow-equations are well defined for any smooth cutoff, no matter how smooth. We tested multiple cutoff-functions (e.g. $f_\lambda=\lambda$ and $f_\lambda=e^{-\lambda}$) and found no difference on the scale of our plots, indicating that the results are essentially independent of the cutoff function. With all tested cutoffs we were able to extract the correct $k$-scaling as $k \to 0$ (i.e. the correct Taylor-coefficients; see fig. 4,5) which is a very sensitive test for the validity and fidelity of the FRG.

The divergence of the flow equations can indeed be avoided by using an appropriate counter-term in the free propagator, this is what Schütz and Kopietz do in their paper (see Ref. [23]: J. Phys. A: Math. Gen. 39 8205). We discuss this briefly below eq. (26) (bottom of page 7). In our truncation schema however, the issue with this approach is that the FRG would resum a different diagrammatic series, viz. a resummation with the free particle ($H_0=\frac{p^2}{2}$) as non-interacting part, which is not desirable.

We hope we could address your comments with this reply. See https://drive.google.com/file/d/16wJ2nPQbA4LBp26msYZBIsOZ2BUf3gcr/view?usp=sharing, for an updated version of the manuscript, where changes and additions (excluding corrected typos) are marked in blue.

Kind regards,
Lukas Grunwald, Volker Meden, Dante Kennes

---

## Round 1 · Referee Report · Anonymous (Referee 2) · 2022-4-20

Strengths

1.) Application of functional renormalization group to non-hermitian systems.
2.) Application of vertex expansion with vacuum expectation values.

Weaknesses

1.) Very technical.
2.) Non-systematic cutoff choice.
3.) No experimental relevance.

Report

In this work the authors use the functional renormalization group
(FRG) to study a simple zero-dimensional toy model for a non-hermitian
quantum system with parity-time symmetry (which guarantees that the spectrum is real).
Starting from the formally exact Wetterich equation (18), the authors use a truncated vertex expansion to calculate the
ground state energy, the first excitation energy, the vacuum expectation value, and the fluctuation of the position operator. A comparison of the FRG results with the exact solution shows that the FRG results are quantitatively accurate even up to large values of the interaction.
The main result of this work is the observation that for a non-hermitian system
a truncated vertex expansion of the FRG
can produce results with a similar accuracy as for a hermitian system.
Although this work is rather technical,
I believe that this result is sufficiently important and interesting to justify publication in SciPost Physics.
However, before this paper can be accepted, the authors should consider the following points:

Requested changes

1.) A slight complication of the non-hermitian toy model considered in this work is that the
position operator $x$ has a finite vacuum expectation value, which implies that the vertex expansion
contains also vertices involving odd powers of the fields. The general form of the FRG vertex expansion
in the presence of vacuum expectation for hermitian systems has already been derived in Ref.~[23], where
the flow of the vacuum expectation value is determined from the condition that the one-point vertex vanishes for any value of the flow parameter. From the discussion after Eqs.(21)-(25) it is not completely clear to me whether the flow equation (22) for the vacuum expectation value is really equivalent to the condition that the one-point vertex vanishes
identically or not.
I suggest that the authors try to present a more transparent discussion of this point in the text around Eqs.(21)-(25).

2.) About the regulator choice: for their explicit calculations
the authors use a simple multiplicative cutoff scheme where the scale-dependent free propagator is
${\cal{G}}_0^{\lambda} ( \omega ) = {\cal{G}} ( \omega ) f_{\lambda} ( \omega )$.
The authors state find for the quantum model, the best results have been obtained for a simple frequency-independent
linear cutoff
$f_{\lambda} ( \omega ) = \lambda$. In footnote 5 the authors state that they have also tested several
frequency dependent cutoffs of the exponential type of the Heavyside type (what is the $x$ in the expressions given
in footnote 5?). The reader gets the impression that the cutoff choice involves a lot of guessing and
there is no systematic way of determining the proper cutoff. But this is not quire correct, because systematic methods of
finding an optimized regulator are available, for example the principle of minimal sensitivity
developed by B. Delamotte and coauthors (Phys. Rev, D 67, 065004 (2003); Phys. Rev. E 95, 012107 (2017)).

3.) The Hamiltonian is defined twice, in Eq.(1) and again in Eq.(2). I think one of these equations
should be deleted.

4.) Printing error in the paragraph after Eq.(8): ``generalize'' should be replaced by ``generalized''.

  • validity: high
  • significance: ok
  • originality: good
  • clarity: good
  • formatting: good
  • grammar: good

Author:  Lukas Grunwald  on 2022-05-05  [id 2443]

(in reply to Report 2 on 2022-04-20)

Dear Referee,
thank you very much for your comments. Regarding the issues you pointed out in your report:

1.) Because of the definition of our Legendre transform eq.(15) the equation of state (which determines $\phi$ for a given $j$) is given by $\Gamma^{(1)} = \frac{\delta \Gamma}{\delta \phi}\Bigr|_{\phi=\phi^\star} = - \big[\mathcal{G}_0 \big]^{-1} \phi^{\star}$ instead of $\Gamma^{(1)}=0$ as in Ref.[23] (We need the modified definition of Legendre transform in order to get well defined initial conditions for the FRG).

Ref.[23] now demands two properties for the free propagator (eq.(26) in Ref.[23]): (I.) $\big[\mathcal{G}_0 \big]^{-1} \phi^{\star} = 0$ and (II.) $Q_\lambda \phi^{\star} = 0$. If we also imply these, we get the same equation of state and indeed then our flow equations for the vacuum expectation value eq.(22) coincides with the one reported in Ref.[23] (eq.(27); note that we need to make the substitution $\mathcal{G}_0 \to -\mathcal{G}_0$ when comparing results between the two papers because of varying definitions). Indeed the Wetterich equations of the two papers coincide in this case, so that all flow equations become equivalent.

From our $\phi^\star$ flow equation eq.(22) we can (analytically) reproduce the correct Taylor coefficients for $\langle x \rangle$ and a comparison with exact numerics, where $\langle x \rangle$ is calculated as $\langle x \rangle = \langle E_0 \vert x \vert E_0 \rangle$, shows good agreement (see fig.3). Thus we think that the two approaches (ours and Ref.[23]) are complementary and become equivalent when the propagator is chosen accordingly. In the new version of the manuscript we added a footnote making the connection between the two approaches more explicit.

2.) Our guiding principle for the cutoff choice (besides fidelity of the results) is numerical stability. We found no cutoff dependence on the scale of the plots (especially: Each tested cutoff correctly reproduced the scaling for $k \to 0$). We have added a small section clarifying this, as well as pointing to the "principle of minimal sensitivity" for a way of choosing an optimized cutoff for a given problem.

3.) We removed the second definition of the Hamiltoinan.

We hope we could address your comments with this reply. See https://drive.google.com/file/d/16wJ2nPQbA4LBp26msYZBIsOZ2BUf3gcr/view?usp=sharing, for an updated version of the manuscript, where changes and additions (excluding corrected typos) are marked in blue.

Kind regards,
Lukas Grunwald, Volker Meden, Dante Kennes

---

## Round 1 · Referee Report · Anonymous (Referee 3) · 2022-4-28

Strengths

1-original work
2-manuscript well written and clear

Weaknesses

1-a more detailed discussion of the two-point vertex would be welcome.

Report

The authors study an exactly solvable PT-symmetric non-linear toy model in the framework of the function renormalization group (FRG). Non-Hermitian systems have already been studied in the past with the FRG but using a derivative expansion (DE) of the effective action. In this work, the authors consider a vertex expansion (VE) of the effective action, a well-known approximation scheme of the FRG approach in the case of Hermitian systems. The main conclusion of the study is that the VE appears to be a valuable method for studying correlation effects in non-Hermitian systems.

The paper is interesting and well written. The study of simple toy models is always an interesting step when assessing a particular method even if the extrapolation to more realistic (and complicated) models is sometimes far from obvious. In the case of the model considered by the authors, the DE is exact in the classical limit, and it is therefore not surprising that both the DE and VE provide us with correct results.

An interesting aspect of the path integral approach is that all particularities due to the non-Hermitian nature of the model are absent, provided that the biorthogonal basis is complete, except for the action being not real. This is pointed out by the authors at the end of Sec.3.1: They emphasize that the modifications of the FRG-VE are not due to the non-Hermitian or PT-symmetric nature of the model but rather to the nonzero expectation value of the field $\phi^*$ (althouth working with a nonzero expectation value of the field is completely standard in the FRG approach; see, e.g., the recent review published in Phys. Rep., 2021).

Before publication, the authors may wish to consider the following comments:

1) A sum over $\omega_4$ is missing in Eq.(10).

2) If the short-hand notation $(j,x)$ means $\int d\tau \, j(\tau) x(\tau)$, shouldn't we have $(j,x)=\sum_\omega j(-\omega) x(\omega)$ (with a minus sign)? The same comment holds for $(j,A\phi)$ and I would expect $j(-\omega)$ instead of $j(\omega)$ in Eq.(16).

3) Since the expectation value $\phi^*=\langle x\rangle$ is complex, I find the notation $\phi^*$ (with a star) slightly confusing since it may be misinterpreted as the complex conjugate (a word of caution might be useful).

4) The meaning of $j(1),\phi(1),...$ in Eqs.(14,17) is not explained.

5) What is the Morris lemma which is alluded to after Eq.(26)?

6) Since one of the advantages of the VE (as compared to the DE) is the possibility to compute the frequency dependence of the vertices, it would be interesting to show the two-point vertex $\Gamma^{(2)}_{\omega,-\omega}$ and, if possible, compare with the exact result. Would it be even possible to extract the spectrum of the model from the analytically continued vertex ($i\omega\to \omega+i0^+$)? At zero temperature, the Padé approximant method is known to be a reliable method to perform the analytic continuation.

7) In the caption of Fig.4, the relation of $\Delta Q$ with the plots is not very clear.

8) Possible typos:

-"frequency depended" (caption of Fig.1, footnote 5 and Sec.4.2). -"We emphasis" (end of Sec.2). -"in which such occurs" (end of Sec.3). -"explicitly time depended nH systems" (conclusion). -"time depended transport" (conclusion). -"The integrals one the right hand side" (Appendix A).

  • validity: high
  • significance: good
  • originality: good
  • clarity: high
  • formatting: perfect
  • grammar: perfect

Author:  Lukas Grunwald  on 2022-05-05  [id 2444]

(in reply to Report 3 on 2022-04-28)
Category:
remark
answer to question

Dear Referee,
thank you very much for your comments. Regarding the issues you pointed out in your report:

2.) The plus-sign in the definitions of $(j, \phi)$ is intentional and chosen for notational convenience. If one couples a source term $\sim j(-\omega) x(\omega)$ this would only lead to several minus-signs in definitions and intermediate steps of the derivation. The same holds for the definition of $(j, A \phi)$. In our notation e.g. the free propagator takes the form $\mathcal{G}(\omega_1, \omega_2) = \mathcal{G}(\omega_1) \delta_{\omega_1 + \omega_2, 0}$. We added a comment on this in the new version of the manuscript.

3.) We added a comment pointing out that the $^\star$ in $\phi^\star$ is part of the symbol and not a complex conjugation.

4.) We changed the notation to $j(\omega_1) \dots j(\omega_n)$ for better clarity.

5.) The Morris lemma is a way to regularize products of the form $\delta(x) h(x, \theta(x))$ that appear, when using the standard-Heaviside cutoff at $T=0$. One can show that $\lim_{\epsilon \to 0} \delta_\epsilon(x) h(x, \theta_\epsilon(x)) = \delta(x) \int_0^1 ds \; h(x, s)$, where $\delta_\epsilon, \theta_\epsilon$ are the broadened Dirac-delta and Heaviside function respectively. For more information on this also see ref [32]: Int. J. Mod. Phys. A 09(14), 2411 (1994).

6.) One plot for $\Gamma^{(2)}_\omega$ can be found in appendix C, fig. 6, where we compare $m_c=2,3$ with the exact results. As to obtaining the complete spectrum: The Lehmann representation of $G^{(2), c}_\omega$ is dominated by the first term, the next term in the sum already being two orders of magnitude smaller. This implies that contribution of higher-order excitations to $G^{(2), c}_\omega$ and $\Gamma^{(2)}_{\omega}$ are very small (i.e. difficult to capture in approximation schemes) such that an analytic continuation $i\omega_n \to \omega + i0^+$ of the FRG results, with e.g. an Padé extrapolation, will not allow an accurate approximation of the full spectrum. We added a remark pointing this out in the new version of the manuscript.

7.) We added a comment in the caption of Fig.4 pointing to the main text, where a more precise definition of $\Delta Q$ is given.

We hope we could address your comments with this reply. See https://drive.google.com/file/d/16wJ2nPQbA4LBp26msYZBIsOZ2BUf3gcr/view?usp=sharing, for an updated version of the manuscript, where changes and additions (excluding corrected typos) are marked in blue.

Kind regards,
Lukas Grunwald, Volker Meden, Dante Kennes

---

## Round 2 · Referee Report · Anonymous (Referee 4) · 2022-5-6

Report

I have read again the manuscript and the reply of the authors to my comments. I think that the authors have properly answered all of my questions and have modified their manuscript accordingly. I now recommend publication of the
manuscript in the present form in SciPost.

---

## Round 2 · Referee Report · Anonymous (Referee 5) · 2022-5-11

Report

The authors have responded convincingly to my comments.
Hence, I now recommend publication of the manuscript in its present form.

---

## Round 2 · Referee Report · Anonymous (Referee 6) · 2022-5-13

Report

I would like to thank the authors for replying to my report. I do not understand the argument as to why it is not possible to obtain the full spectrum using Padé approximants to perform the analytic continuation. For a zero-space dimensional problem this is quite simple to do and this would really have improved the paper. On the other hand I do not want to further delay the publication and I therefore recommend publication of the manuscript in its present form.

  • validity: -
  • significance: -
  • originality: -
  • clarity: -
  • formatting: -
  • grammar: -

Author:  Lukas Grunwald  on 2022-05-19  [id 2497]

(in reply to Report 3 on 2022-05-13)

Dear Referee,
thank you very much for your comment.
Pade-extrapolations are very sensitive to the details of the ω-grid as well as numerical/statistical noise, leading to inaccuracies in the final result (we previously tested the Pade-extrapolations for the anharmonic oscillator in the context of Ref. [26]). Since higher order excitations only have a very small weight in the Greens function (previous argument), a reliable extraction of excitation energies will not be possible with this approach, due to the inaccuracies of the extrapolation.

We hope this further clarifies our argument given in the paper.

Kind regards,
Lukas Grunwald, Volker Meden, Dante Kennes

---

## Round 2 · Author Response

See https://drive.google.com/file/d/16wJ2nPQbA4LBp26msYZBIsOZ2BUf3gcr/view?usp=sharing, for an updated version of the manuscript, where changes and additions (excluding corrected typos) are marked in blue.

---

## Round 2 · List of Changes

Comments from referee 1:
- Added clarification, that we assume non degenerate eigenvalues only for notational convenience

Comments from referee 2:
- Added footnote clarifying the connection between our approach and the one used by Schütz and Kopietz ( J. Phys. A: Math. Gen. 39 8205)
- Added remark outlining our reasoning for the cutoff choice as well as a reference to the "principle of minimal sensitivity"
- Removed second definition of the Hamiltonian

Comments from referee 3:
- Added footnote explaining sign convention in (j,ϕ)
- Added a footnote pointing out that the in ϕ is part of the symbol and not a complex conjugation
- Changed notation in definition of Greens and Vertex functions for better clarity
- Added footnote explaining why using a numerical analytic continuation will not allow an accurate approximation of the full spectrum
- Added a comment in the caption of Fig.4 for a more precise definition of ΔQ

Additionally we corrected typos in the new version of the manuscript.

---

## Editorial Decision

published